# Effect of Temperature on the Early Development of *Paralichthys olivaceus* Otoliths

**DOI:** 10.3390/ani15060814

**Published:** 2025-03-13

**Authors:** Gao Meng, Jiabao Tang, Qinglin Wang, Zhaohui Sun, Shanshan Yu, Fei Si

**Affiliations:** 1Hebei Key Laboratory of the Bohai Sea Fish Germplasm Resources Conservation and Utilization, Beidaihe Central Experiment Station, Chinese Academy of Fishery Sciences, Qinhuangdao 066100, China; mengg@bces.ac.cn (G.M.); 17686268285@163.com (J.T.); sunzh@bces.ac.cn (Z.S.); 2Bohai Sea Fishery Research Center, Chinese Academy of Fishery Science, Qinhuangdao 066100, China; 3National Agricultural Experimental Station for Fishery Resources and Environment Qinhuangdao, Qinhuangdao 066100, China; 4College of Marine Resources & Environment, Hebei Normal University of Science & Technology, Qinhuangdao 066000, China; wangqinglin_1984@163.com; 5Hebei Key Laboratory for Ocean Dynamics, Resources and Environments, Qinhuangdao 066000, China

**Keywords:** *Paralichthys olivaceus*, otoliths, development, temperature

## Abstract

This study aimed to understand how water temperature changes affect early growth patterns in otoliths of *Paralichthys olivaceus* (hereafter *P. olivaceus*). Natural temperature variations (15–19.5 °C) were compared with constant 20 °C (±0.5 °C) conditions during the first 50 days of fish development. The results revealed that (1) daily growth rings in otoliths formed five days after hatching under both temperature conditions; (2) fish in natural temperatures showed faster otolith growth after 35 days and clearer growth ring patterns compared to those in constant temperatures; (3) otolith size consistently correlated with fish body length across all groups. These findings will improve methods for tracking fish growth in changing environments. By demonstrating that natural temperature fluctuations create more readable growth records in otoliths, this study helps scientists better reconstruct wild fish life histories and optimise breeding programmes for species conservation, ultimately supporting sustainable seafood production and ocean ecosystem management.

## 1. Introduction

The otolith is a calcified structure located in the inner ear of teleost fishes, consisting of 99% calcium carbonate. It has a stable morphological structure and chemical composition, preserving extensive biological, physical, and chemical information [1,2,3]. Examining the growth, development, and microstructure of otoliths provides insight into the life history of fish and the environmental changes they have experienced, facilitating age determination and population classification. For example, fish otoliths correlate closely with growth and environmental factors, with their growth rate closely reflecting overall fish growth [4,5]. Levels of strontium (Sr), calcium (Ca), and other trace elements in otoliths vary with salinity and other environmental factors [6], making them critical tools for analysing fish growth history and their surrounding environmental conditions. These features provide a reliable basis for in-depth studies on the biological characteristics and ecological adaptations of fish.

The early morphological changes in fish otoliths are similar, but their developmental characteristics and whorl deposition patterns exhibit species-specific traits influenced by environmental and other factors. Consequently, the impact of environmental conditions on early otolith development has attracted considerable global research interest. Research has shown that otolith growth is regulated by the central nervous system, genetic factors, and environmental conditions, including salinity, photoperiod, and temperature. Salinity variations affect the incorporation of trace elements like Sr and Ca into otoliths [6,7]. Photoperiod primarily influences otolith growth and development by modulating fish biorhythms and hormone secretion [8]. Temperature is among the most critical factors affecting otolith growth and whorl formation [9,10]. The growth rate and morphology of otoliths undergo substantial changes in response to temperature fluctuations. In 2020, Dariusz et al. reported that elevated temperatures significantly enhanced otolith growth in Sphyraena spp. under experimental conditions [11]. Similarly, in 2022, George et al. demonstrated that a 3 °C temperature increase during the embryonic and yolk-sac larval stages of *Sparus aurata* significantly modified otolith shape and bilateral symmetry [12]. Temperature also profoundly impacts otolith microstructures, as higher temperatures accelerate otolith growth, producing broader and looser structures, while lower temperatures slow growth, resulting in narrower and denser formations [13].

*P. olivaceus*, widely distributed in the Yellow and Bohai Seas, is considered a key species for marine aquaculture enhancement in East Asia [14]. In recent years, wild populations of *P. olivaceus* have significantly declined due to factors such as marine environmental pollution [15]. To address this, East Asian countries have implemented juvenile release programmes since the 1980s to replenish natural populations. Japan began releasing *P. olivaceus* in 1988–1989, releasing over 100,000 individuals during this period [16]. Since 2007, China has consistently undertaken stocking and releasing activities in the Bohai Sea, while South Korea released over 5 million individuals between 2000 and 2008.

Despite these efforts, the artificial breeding of *P. olivaceus* seedlings is conducted at temperatures exceeding those of natural waters, and limited studies exist on the impact of temperature changes on the early development of *P. olivaceus* otoliths. Specifically, the effects of temperature on otolith growth patterns, morphological differences and potential implications for population structure are poorly understood. This study examines the developmental characteristics of *P. olivaceus* otoliths during early growth stages under varying temperature conditions. By analysing morphology, whorl formation, and microstructural traits, this study aimed to elucidate the effects of temperature on otolith development. The findings will establish a foundation for reconstructing the early life history of *P. olivaceus* through otolith analysis, which will contribute to the conservation of marine fish stocks and the sustainable management of resources.

## 2. Materials and Methods

### 2.1. Sample Collection

The experimental materials consisted of artificially fertilised eggs from wild-caught *P. olivaceus* broodstock maintained at the Beidaihe Central Experiment Station at the Chinese Academy of Fishery Sciences. The fertilised eggs were assigned to two groups. The first group was incubated in a 300 L blue water tank at natural water temperatures (15–19.5 °C) in a flow-through water system. This was the actual temperature variation range of breeding water throughout the experimental period. A blower connected to an air stone ensured sufficient oxygen, and the sieve mesh was rinsed daily in the evening to prevent water flow obstruction caused by mesh blockage. The second group was placed in a static water environment, with the temperature controlled at 20 ± 0.5 °C. The selection of 20 °C was based on this temperature lying within the optimal growth temperature range (16–22 °C) for *P*. *olivaceus*, representing idealised aquaculture conditions. Furthermore, studies have demonstrated that 20 °C optimises metabolic efficiency and energy utilisation in *P. olivaceus* compared to lower temperatures [17]. Heating rods maintained the temperature, and oxygenation was from air stones. Water changes (each replacing 90% of the water volume) were performed twice a day, with one synchronised with siphoning of bottom sediments once a day.

The initial feed consisted of rotifers, followed by twice-daily feedings of an appropriate amount of rotifers. Starting on day 14, brine shrimp nauplii and a specified amount of algal suspension were introduced four times daily to sustain nauplii activity. Beginning on the first day after hatching, 15 larvae were randomly selected each day for 30 days. Samples were taken every five days from day 35 to day 50 after hatching.

### 2.2. Processing of Samples

Once the samples were collected, the fish were positioned on a slide, dried using filter paper, and photographed with an optical microscope. Their total length (*TL*) was measured with system software accurate to 0.01 mm. The fish were then labelled and placed in centrifuge tubes with anhydrous ethanol. After the photographs were taken, the sagittae and lapillus were extracted using a dissecting microscope, their organic films were removed, and they were washed with anhydrous ethanol, and then dried and secured with neutral tree glue for future examination. Otoliths were identified based on their anatomical position within the inner ear structure, with the sagitta located in the utricle and the lapillus in the saccule. Only left otoliths were used for analysis to ensure consistency. Paired otoliths were collected from randomly selected specimens. Otoliths that were challenging to identify or had damage from light refraction or cracks were not included. Unprocessed samples were kept in 1.5 mL centrifuge tubes containing anhydrous ethanol.

During the first 30 days, the otoliths did not require processing as the daily rings were visible. After 30 days, sandpaper grit sizes (2000, 3000, and 5000 grit) were selected to progressively refine sagittal otolith surfaces for microstructural analysis. The samples were continuously observed under a microscope during grinding to avoid over-grinding. The ground otoliths were cleaned with water to eliminate surface powder, dried with mirror paper to remove water stains, and photographed under a microscope (Leica Microsystems Ltd, Wetzlar, Germany. DFC7000T&M165FC). Motic Images Plus 2.0 software was used to measure the long and short axes of the otoliths. The simulated natural water temperature mode *P. olivaceus* larvae designated as *N*, and the constant temperature mode was designated as R. Due to the delayed appearance of asterisci compared to sagittae and lapillus, as well as the irregularity of their daily rings, these daily rings were not analysed.

### 2.3. Data Analysis and Statistics

The otolith morphology development images were edited and processed with Adobe Photoshop 2022, ensuring the otolith characteristics remained unchanged. The relationship between *TL* and days post-hatching (*D*) was modelled using the power function. The results are presented as mean ± standard deviation (mean ± SD), and differences were analysed using an independent sample *t*-test in SPSS Statistics 26.0. A *P*-value less than 0.05 was considered significant, whereas a *P*-value less than 0.01 was considered highly significant. The images were analysed and processed using Origin 2022.

## 3. Results

### 3.1. Effect of Temperature on the Microstructure of Otoliths

The timing of the initial daily ring in *P. olivaceus* otoliths was not influenced by the two temperature modes. From 0 to 4 days after hatching, no clear or consistent daily rings were seen on the otolith surfaces. On the fifth day following hatching, a ring became apparent, and each subsequent day saw the formation of another ring. The equation *D* = *N* + 5 describes the connection between the age of the young and the count of daily rings on the otolith. Each full daily ring comprised a broader growth band and a thinner intermittent band, with the growth band being clear and the intermittent band having a dull colour. The colour of the first-day pairs of otoliths (lapillus, sagittal otoliths) was significantly darker than the others, and at ages 2–4, dark bands of relatively lighter colour were interspersed between the two darker bands. As the otoliths aged, the daily rings in the simulated natural water temperature mode became clearer than those in the thermostatic mode. Figure 1 presents the otolith morphology of *P. olivaceus* at three developmental stages: 10, 20, and 30 days post-hatching (dph).

### 3.2. Temperature Effects on the Growth of Juvenile Fish

The results showed a positive correlation between age (day) and fish length in both temperature modes. During the experimental period, the *P. olivaceus* samples ranged from 2.8 mm–28.67 mm in total length. Sampling was divided into two stages. In stage one (0–30 days), the average growth rate was 0.2734 mm·d^–1^ in the thermostatic mode and 0.1848 mm·d^–1^ in the simulated natural water temperature mode, with no significant difference in growth rates between modes (*P* > 0.05). In stage two (35–50 days), the average growth rate in the natural water temperature mode was 0.5863 mm·d^–1^. The average growth rate in the constant temperature mode was 0.3679 mm·d^–1^, significantly lower than in the simulated natural water temperature mode (*P* < 0.05; Table 1). During the whole sampling period, the relationship between total length (*TL*) and days old (*D*) for young fish was *N_TL_* = 0.4248*D* + 0.6221 (*R*^2^ = 0.9444) for the natural water temperature mode and *R_TL_* = 0.3585*D* + 2.4951 (*R*^2^ = 0.9400) for the constant temperature mode (Figure 2). The temperature profiles of the water environments in the natural water temperature mode during the experimental period, as shown in Figure 3.

### 3.3. Effect of Temperature on the Time of Otolith Formation and Morphological Changes

Temperature differences did not influence the formation timing of sagittal otoliths and lapillus, with both pairs forming simultaneously at the time of egg fertilisation. When *P. olivaceus* sagittal otoliths and lapillus were subrounded (Figure 4a). During subsequent development, the lapillus exhibited a consistent overall developmental trend under both temperature conditions, with their cores shifting from subrounded to mussel-shaped between 20 and 25 dph (Figure 4b). They continued to develop beyond this stage, with significant morphological differences observed by the end of the sampling period.

Under the *R* mode, sagittal otoliths began extending at both ends at 25 dph, transitioning from subrounded to oval (Figure 4c). Their growth was steady, with smoother edges and initially complex internal structures. By day 30, some otoliths became pear- or arrow-shaped (Figure 4d,e), showing accelerated growth with slightly pointed tops and clearer rings. By the end of the sampling period, their tops had rounded, growth slowed, and their shapes became regular with homogeneous rings.

Under the *N* mode, at 25 dph, the otoliths initially remained nearly round before transitioning to a rounded square with petal-like sides (Figure 4f) and eventually to an oval shape, undergoing a more intricate developmental process. By day 30, some otoliths also became pear- or arrow-shaped. At the end of sampling the otolith tops were similarly rounded and smooth. However, although their developmental pathways diverged in the early stages, resulting in distinct morphological differences from compared to the *R* group.

### 3.4. Effect of Temperature on Otolith Growth and Development and Microstructure

#### 3.4.1. Effects of Different Temperature Modes on the Growth of Sagittal Otoliths

Under both temperature modes, sagittal otolith growth increased with age, with an exponential relationship between the long and short axes and age. The relationship equations were as follows: *R_Ls_* = 0.0242e^0.0675*D*^ (*R*^2^ = 0.9379) and *R_Ws_* = 0.0169e^0.0718*D*^ (*R*^2^ = 0.9379) in the thermostatic mode; *N_Ls_* = 0.0026e^0.1113*D*^ (*R*^2^ = 0.8889) and *N_Ws_* = 0.0141e^0.0660*D*^ (*R*^2^ = 0.9494) in the simulated natural water temperature mode (Figure 5a,b). The average otolith rate was calculated using 30 days as the cut-off point. Before 30 days, the average growth rate of the sagittal otolith long axis was 0.0022 mm·d^–1^ in the simulated natural water temperature mode and 0.0058 mm·d^–1^ in the thermostatic mode; after 30 days, these rates increased to 0.0195 and 0.0251 mm·d^–1^, respectively. The two phases of sampling were analysed using the t-test for independent samples. This revealed that the average growth rates of the long and short sagittae axes were significantly different (*P* < 0.05) under the thermostatic mode before and after sampling, while there was no significant difference (*P* > 0.05) under the simulated natural water temperature mode (Table 2). The long and short axes of sagittal otoliths in the simulated natural water temperature mode grew nearly isometrically, with *N_Ws_* = 0.9193*L_s_* (*R*^2^ = 0.9984), whereas the long axis of sagittal otoliths in the thermostatic mode grew a little faster, with *R_Ws_* = 0.7819*_Ls_* (*R*^2^ = 0.9890) (Figure 5c).

The growth of otoliths was positively correlated with *TL* (Figure 5d,e). Because the long axis of the sagittal otoliths was exponentially related to the *TL*, the difference in otolith size increased as the *TL* increased. A differential analysis of the two temperature patterns of sagittal otoliths revealed that when the *TL* was constant, the long and short axes of sagittal otoliths of the two temperature patterns were significantly different (*P* < 0.05).

#### 3.4.2. Effect of Different Temperature Patterns on Lapillus Growth

Under the different temperature regimes, lapillus growth exhibited an exponential relationship with dph, with size differences emerging as the dph increased (Figure 6a,b). Notably, significant differences (Table 3) (*P* < 0.05) were observed in the short axis of lapillus. However, temperature did not impact near-isometric lapillus growth (Figure 6c).

The growth of otoliths was positively correlated with total length. The long and short axes of lapillus were exponentially correlated with total length. Otoliths in the thermostatic mode were generally larger under the same total length condition. A statistical analysis of sagittal otolith morphology under two temperature regimes revealed significant differences in long and short axis dimensions at equivalent body lengths (Figure 6d,e).

## 4. Discussion

The determination of the first day of otolith whorl formation and the continuity and completeness of whorl formation are particularly important in the age determination of juvenile fish. In this study, the first whorl of the *P. olivaceus* juveniles formed 5 days after the emergence of membranes in both the sagittae and lapillus under the two temperature modes. Growth accelerated, but otolith ring deposition remained consistent at once per day, demonstrating cyclic daily ring formation. However, prior studies indicate that suboptimal temperatures may inhibit or disrupt daily ring formation, potentially leading to age underestimation [18]. The natural water temperatures modelled in this study were below the thermostatic regime, but still within the survival range of *P. olivaceus*. This did not affect the timing of exogenous nutrient uptake. Consequently, there was no difference in the time of first appearance of diurnal rings between temperature patterns.

The lapillus of *P. olivaceus* in both temperature modes showed essentially the same trend in shape change, from subround to shell-like. However, the sagittal otolith shape-change process showed different trends under the different temperature modes. In the simulated natural water temperature mode, sagittal otoliths developed more gradually, passing through an intermediate stage between round and square before transitioning from an elliptical to an arrow-shaped form. By the end of the sampling period, only a subset of otoliths had attained a pear-shaped or arrow-shaped form. In contrast, the present study found that temperature affects the size of otoliths in early *P. olivaceus*, but does not affect otolith morphology, contrary to the findings in the aforementioned studies.

The reasons for the differences are twofold: first, the samples were collected at different times between the two studies, with the otolith morphology of *P. olivaceus* stabilising 60 days after hatching; second, otolith shapes were influenced by a complex interplay of endogenous and exogenous factors, encompassing abiotic environmental parameters (e.g., temperature, CO_2_ concentration) and biotic parameters (e.g., food availability). In contrast, otolith morphology in early *P. olivaceus* is primarily influenced by genetic rather than environmental factors, resulting in negligible morphological differences.

Early *P. olivaceus* otoliths were round or subround, especially lapillus with almost equal growth of the long and short axes, which did not have a significant effect on the shape of otoliths [19,20]. Examining trend plots depicting the relationship between otolith size and day-age under varying temperature regimes revealed temperature significantly influences otolith size. Furthermore, temperature significantly impacts *P. olivaceus* larvae growth and microstructure holding practical importance for age identification using otoliths, understanding early life history, estimating growth rates, and comprehending environmental change effects on biological populations. Otolith use holds practical significance. Fish morphology is typically influenced by genetic factors, environmental conditions, and individual physiology. Genetic differences accumulate over time, making morphological characteristics more indicative of geographic group differences [21].

In the simulated natural water temperature mode, the initial incubation temperature was low, which limited the feeding ability of *P. olivaceus* under suboptimal conditions. Consequently, the availability of essential nutrients for growth was restricted, resulting in slower fish growth. The growth of otoliths was closely related to the growth of the fish, so the otolith growth rate was also reduced. In contrast, fish exhibited enhanced growth at the optimal temperature of 20 °C, leading to the formation of larger otoliths than those observed in the simulated natural water temperature mode [22,23,24]. Initially, fish growth rates in the simulated natural water temperature mode were slower than those in the constant temperature mode for the first 30 days. However, this trend reversed thereafter, possibly due to the simulated natural water temperature reaching the optimal range for *P. olivaceus* growth. There is research indicating that suitable temperature changes enhance digestive enzyme activity such as ALT and AST, promote protein metabolism, and improve nutrient digestion and absorption in fish. Conversely, unsuitable temperatures can harm organisms [25,26]. Therefore, optimal temperatures increase fish growth rates, influencing otolith growth and development. Otolith size correlates with fish body length [27,28,29]. A comparison between the two temperature modes showed larger sagittal and lapillus growth in the constant temperature mode compared to the simulated natural water temperature mode.

Temperature fluctuations during otolith growth and development can significantly affect otolith clarity. Lei et al. [30] compared the otolith microstructures of field-collected and captive-reared coelacanths. They observed that otolith growth rings of fish larvae reared in variable-temperature conditions were clear, whereas those reared in constant temperatures were fuzzy or unrecognisable. Yan et al. [31] investigated temperature effects in the release tagging of Cannabis sativa. They demonstrated that regular temperature fluctuations can induce regular changes in brightness, darkness, width, and narrowness of otolith daily rings, with clearer patterns observed under variable temperatures compared to constant temperatures. Specific temperature fluctuations can promote daily ring formation, enhancing clarity and contrast between daily ring. This study’s findings align with previous research, indicating clearer daily rings under natural conditions. In contrast, those raised in constant 20 °C temperatures exhibited unclear and poorly defined otolith daily rings. Furthermore, the clarity of daily rings varies among otoliths from different origins. Song et al. [32] compared otolith daily rings of grass carp larvae in natural and cultured environments. They observed clearer daily rings in cultured fish, whereas those in the wild exhibited unclear and indistinct boundaries. Transferring wild fish to cultured environments and continued feeding resulted in distinctly striped otolith daily rings.

In terms of ring deposition, otoliths from the natural water temperature mode exhibited clearer and darker daily rings compared to those from the constant temperature mode, with a more pronounced effect observed under greater temperature fluctuations. The influence of temperature on otolith deposition and daily ring formation primarily arises from its impact on organismal metabolism [33]. Ge et al. noted unclear rings contrasts and indistinct boundaries in fish reared in constant temperatures, sometimes rendering daily rings invisible, whereas day–night temperature variations promote distinct light–dark ring patterns [34]. The degree of temperature difference in water correlates with rings contrast clarity greater differences enhance visibility, aiding in age determination. This underlies preconditioning cultured fish with periodic temperature fluctuations of at least 4 °C before release, altering ring deposition patterns in otoliths compared to those in fish reared under constant temperatures. This marking method is widely employed for fry release, offering advantages over fluorescent, molecular, and elemental [35,36,37] marking due to high success rate, cost-effectiveness, minimal impact on fry, and superior legibility. It also enables the differentiation of released fish across watersheds based on varying warming cycles.

Otoliths are calcium carbonate structures formed through natural biomineralisation in the inner ears of fish, with their microstructures serving as indicators of biological information such as fish age [38,39,40]. Changes in fish physiological responses to altered water environments result in distinct daily markings on otoliths. Therefore, studying early temperature effects on otolith microstructures, understanding otolith growth patterns, and assessing temperature impacts on otolith microstructures facilitate otolith use in population identification. This approach holds significant implications for accurately understanding early fish life histories, environmental changes, and early population dynamics, and serves as a crucial scientific foundation for fishery resource investigations.

## 5. Conclusions

Variations in culture water temperature influenced the deposition of otolith daily rings in *P. olivaceus*. Otolith ring deposition was more distinct at natural water temperatures (15–19.5 °C) than at a constant temperature of 20 °C (±0.5 °C), although the first daily ring appeared at approximately the same time under both conditions. The relationship between the long (*L*) and short (*S*) axes of sagittal otoliths and total length followed exponential functions in both temperature modes: *R_Ls_* = 0.0109e^0.2048*TL*^ (*R*^2^ = 0.9440), *N_Ls_* = 0.0142e^0.1537*T*L^ (*R*^2^ = 0.9732); *R_Ws_* = 0.0113e^0.1941*TL*^ (*R*^2^ = 0.9350), *N_Ws_* = 0.0151e^0.1544*TL*^ (*R*^2^ = 0.9732). Lapillus exhibited a power function relationship with total length: *R_Ll_* = 0.002*TL*^1.5082^ (*R*^2^ = 0.8943), *N_Ll_* = 0.0068*TL*^0.9546^ (*R*^2^ = 0.9129), *R_Wl_* = 0.002*TL*^1.4833^ (*R*^2^ = 0.9126), *N_Wl_* = 0.0063*TL*^0.9521^ (*R*^2^ = 0.9028). Otolith growth rates accelerated after 30 days compared to earlier stages. Significant differences (*P* < 0.05) were observed in the short axes of sagittal otoliths and highly significant differences (*P* < 0.01) in the short axes of lapillus in the thermostatic mode, whereas otoliths in simulated natural water temperatures did not show significant differences (*P* > 0.05). Although otolith growth rates were faster in constant temperature mode compared to simulated natural water temperature mode, the difference was not statistically significant (*P* > 0.05).

## Figures and Tables

**Figure 1 animals-15-00814-f001:**
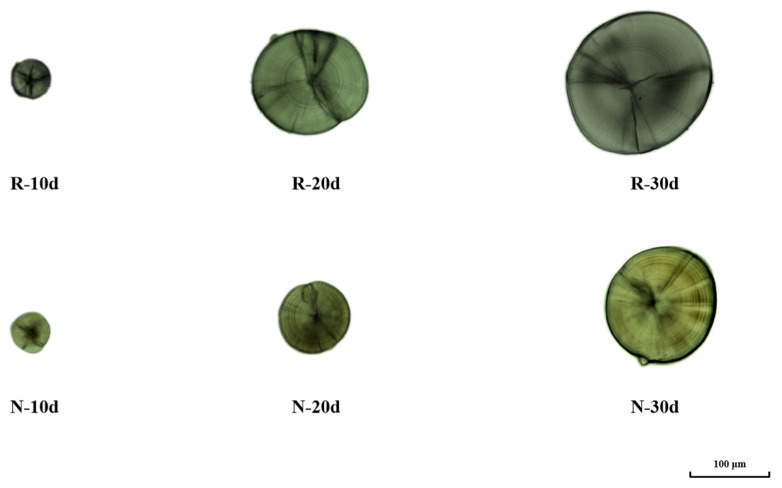
Otoliths of *P. olivaceus* at different developmental stages (10, 20, and 30 days old).

**Figure 2 animals-15-00814-f002:**
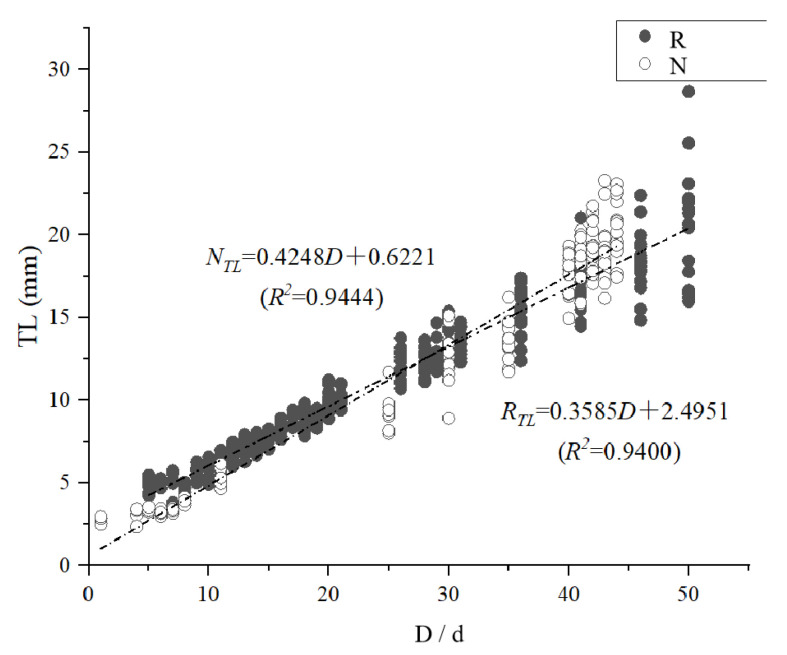
Relationship between total length (*TL*) and days old (*D*) of *P*. *olivaceus* at different temperatures.

**Figure 3 animals-15-00814-f003:**
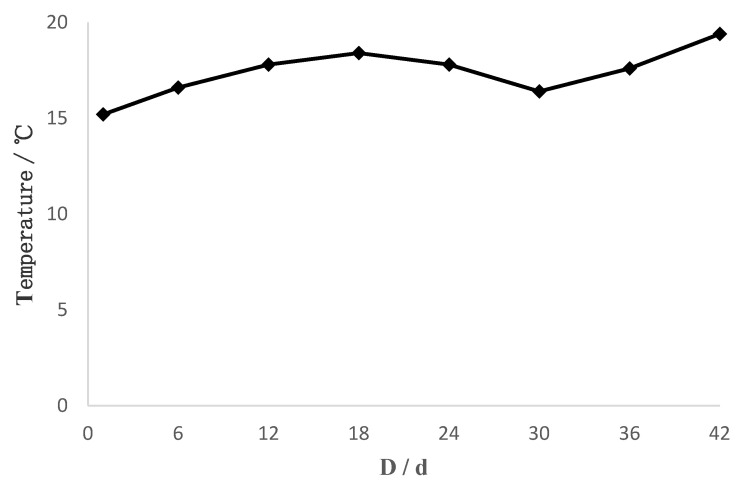
Temperature dynamics of the *N* group.

**Figure 4 animals-15-00814-f004:**
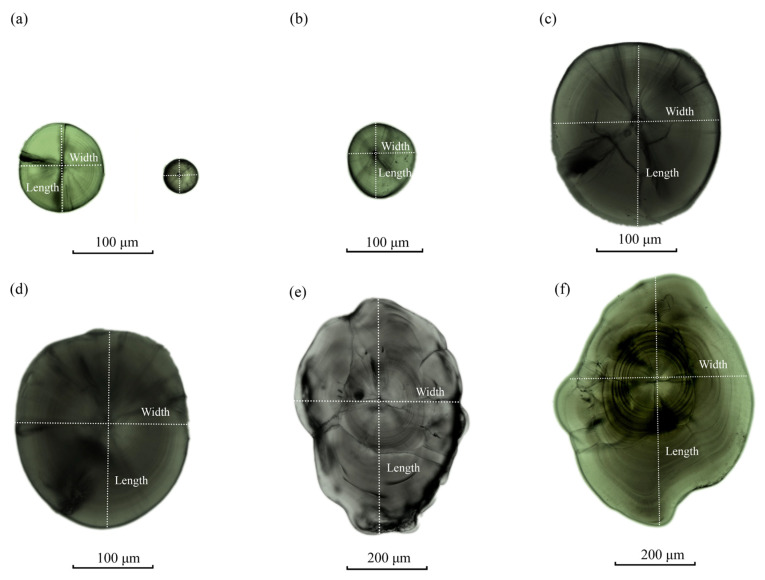
Ontogenetic morphological variations in otoliths in *P. olivaceus.* Note: (**a**) subrounded otoliths—sagittal otoliths (**left**), lapillus (**right**); (**b**) mussel-shaped lapillus; (**c**) oval sagittal otolith; (**d**) petaloidal-sagittal otolith; (**e**) sagittal otoliths—pyriform; (**f**) sagittal otoliths—arrow-shaped.

**Figure 5 animals-15-00814-f005:**
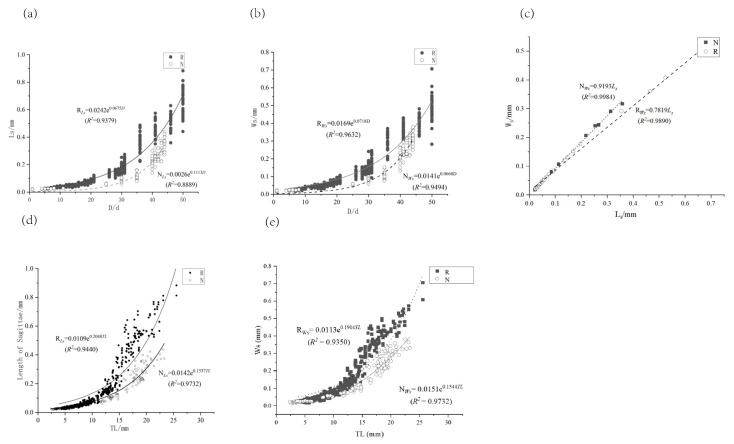
Correlation analysis between *Ls*, *Ws*, *D*, and *TL* of sagittal otoliths in two temperature models. Note: (**a**) Relationship between sagittal otolith long axis (*L_s_*) and day age (*D*) in different temperature patterns; (**b**) Relationship between short axis of sagittal otoliths (*W_s_*) and day age (*D*) in different temperature patterns; (**c**) Relationship between short (*W_s_*) and long (*L_s_*) axes of sagittal otoliths in different temperature modes; (**d**) The relationship between length of sagittae and total length of different temperature; (**e**) The relationship between width of sagittae and total length of different temperature.

**Figure 6 animals-15-00814-f006:**
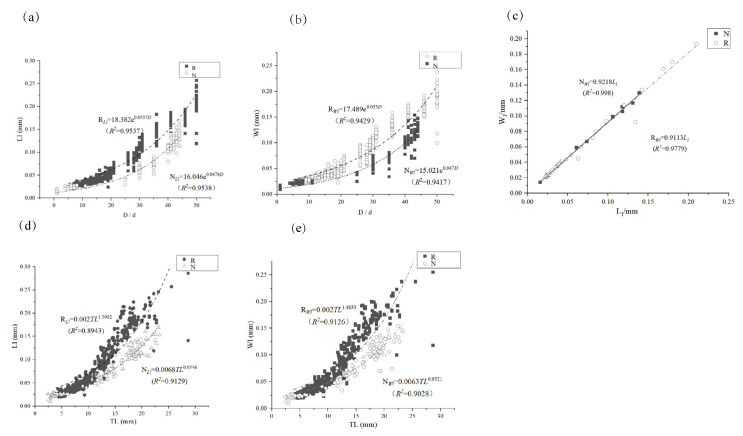
Correlation analysis between *Ls*, *Ws*, *D*, and *TL* of lapillus in two temperature modes. Note: (**a**) The relationship between length of lapillus and age of different temperature; (**b**) The relationship between width of lapillus and age of different temperature; (**c**) The relationship between width and length of lapillus of different temperature; (**d**) The relationship between length of lapillus and total length of different temperature; (**e**) The relationship between width of lapillus and total length of different temperature.

**Table 1 animals-15-00814-t001:** Independent sample t-test for the mean growth rate of *P. olivaceus* at two sampling stages under two temperature modes.

Time	Group	Average	*T*	*P*
0–30 d	*N*	0.1848	2.274	0.598
	*R*	0.2734		
35–50 d	*N*	0.5863	0.863	0.039 *
	*R*	0.3679		

Note: * indicates significant difference (*P* < 0.05).

**Table 2 animals-15-00814-t002:** Independent sample t-test for mean growth rate of sagittal otoliths at two sampling stages in two temperature modes.

	Group	Time	Average	T	*P*
*L_s_*	*N*	0–30 d	0.0017	−13.827	0.182
		35–50 d	0.0162		
	*R*	0–30 d	0.0030	−21.308	0.047 *
		35–50 d	0.0262		
*W_s_*	*N*	0–30 d	0.0016	−15.490	0.321
		35–50 d	0.0147		
	*R*	0–30 d	0.0029	−17.348	0.015 *
		35–50 d	0.0187		

Note: * indicates significant difference (*P* < 0.05).

**Table 3 animals-15-00814-t003:** Independent sample t-test for mean growth rate of lapillus at two sampling stages under two temperature conditions.

	Group	Time	Average	T	*P*
*L_l_*	*N*	0–30 d	0.0056	−11.847	0.614
		35–50 d	0.0043		
	*R*	0–30 d	0.0022	−21.308	0.627
		35–50 d	0.0284		
*W_l_*	*N*	0–30 d	0.0014	−11.91	0.485
		35–50 d	0.0045		
	*R*	0–30 d	0.0019	−17.348	0.009 **
		35–50 d	0.0058		

Note: ** indicates highly significant difference (*P* < 0.01).

## Data Availability

The original contributions presented in this study are included in the article. Further inquiries can be directed to the corresponding authors.

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
