# Peer review of "Effect of Temperature on the Early Development of Paralichthys olivaceus Otoliths"

_animals, 2025, doi:10.3390/ani15060814_

Round 1

Reviewer 1 Report

Comments and Suggestions for Authors

There are still not so many researches focusing on the influence of temperature to otolith shape of fish at early life history. The authors tried to provide the comparative information by an otolith case of olive flounder using two temperature condition of natural 15°C–19.5°C and constant 20°C, respectively, during the first 50 days of development. I just want to provide some minor corrections for the manuscript.

(1) Why the natural temperature varied 15°C–19.5°C? It was natural daily temperature cycle of water in experimental tank? How could the authors control the water temperature at 20±0.5℃ in another experimental tank?

(2) Lapillus and Sagitta were used in this study, but why Asteriscus were not selected? What will be happened for this type of otolith?

(3) Please combine Figs. 3-8 into a new integrated figure and each former figure could be as corresponding subfigure (a), (b), (c), …… in it. The same work is necessary for Figs. 9-13, and Figs. 14-18.

(4) Please follow the article formation of Animals throughout the manuscript, e.g., that for numbers of literature in text should be [1, 2, 3], and can not be presented as 123. All scientific name of species and statistical symbols should be in italic fonts.

(5) Full name of Paralichthys olivaceus needs only to be given at the earliest place to appear in text, and then, given as P. olivaceus.

Comments on the Quality of English Language

I feel that the English could be improved to more clearly express the research.

Reviewer 2 Report

Comments and Suggestions for Authors

This study investigates how temperature regimes (natural fluctuating: 15–19.5°C vs. constant: 20°C±0.5°C) influence the early growth, microstructure, and ontogeny of otoliths in Paralichthys olivaceus (brown flounder).

The study highlights the potential of otolith microstructure as a tool for reconstructing early life histories and optimizing stocking strategies for P. olivaceus. However, by excluding intermediate or extreme temperatures, the study fails to address critical thresholds for otolith development.  For instance, temperatures above 20°C might accelerate growth but compromise structural integrity, while temperatures below 15°C could delay otolith formation.  This narrow scope limits the applicability of findings to broader environmental contexts.

Comments on the Quality of English Language

The English could be improved to more clearly express the research.

Reviewer 3 Report

Comments and Suggestions for Authors

Dear Authors,

The study addressed an original topic, and the methods applied were up-to-date. Statistical applications were clearly stated. There are some deficiencies in terms of spelling and language, and these corrections are shown in the pdf file. The following points should be noted:
-In this study focusing on otoliths, it should be stated whether otoliths were removed in pairs and whether right and left distinctions were made.
-Measurements taken from otoliths (short and long axes) should be shown in a figure with the directions of the corresponding part of the otolith.
-The indications of the references in the text are incorrect. For example, age4042.
-In the discussion section, more attention should be paid to the studies conducted on the otoliths of the species.

Other corrections are available in the attached file.

Reviewer 4 Report

Comments and Suggestions for Authors

The abstract is well-structured, the introduction provides a comprehensive background on life-history fish studies in liaison with the effect of environmental parameters. The study significantly contributes to the gap of knowledge, regarding the effect of temperature on Paralichthys olivaceus growth, especially in early life stages. The experimental design is robust with detailed experimental description ensuring reproducibility of the study. Statistical analysis used are the appropriate ones. The results are presented clearly and figures - tables are effectively illustrated the key findings of the study. The discussion section is coherent with justified examples from the literature regarding the fish growth in relation to environmental changes and the implication for fisheries management and conservation. It is also important the inclusion of limitations of the study, including the lack of data on physiological functions. The conclusion succinctly summarizes the key findings. Reference included update a wide range of updated peer-reviewed studies.

A minor edit is that it could be useful for the audience to understand the choice of the constant temperature of 20°C, thus the authors need to provide, based on literature, an explanation for this choice.
